# The Pedagogical Leadership of the Mathematics Faculty: A Systematic Review

**Inmaculada García-Martínez** [1] **, José Luis Ubago-Jiménez** [2,*] **, Jesús López-Burgos** [1] **and Pedro Tadeu** [3]

1   Department of Didactics and School Organization, University of Granada, 18071 Granada, Spain; igmartinez@ugr.es (I.G.-M.); jesus93lb93@gmail.com (J.L.-B.)
2   Department of Didactic of Musical, Plastic and Corporal Expression, University of Granada, 18071 Granada, Spain
3   Research Unit for Inland Development (UDI)—Polytechnic of Guarda, 6300-559 Guarda, Portugal; ptadeu@ipg.pt
*   Correspondence: jlubago@ugr.es; Tel.: +34-958-246-685

**Abstract:** Background: Research on educational leadership has transcended the international sphere. Numerous studies have been developed on this factor of educational improvement. Few is their number, contextualized in the mathematics area and specifically the teachers. Methods: This paper presents a systematic review that highlights the importance of school leadership and mathematics education, providing empirical evidence on the positive impact that the former has on the latter. The method has been adapted to the guidelines promulgated in the PRISMA declaration, to ensure its systematicity. Results: Regarding the results, most of the research included in this review has found positive leadership effects on teacher professionalism, teaching and learning processes, and student performance. Conclusions: As limitations, the prescriptive nature of legislation and organizational structures has been found, which impedes the implementation of more effective leadership modalities.

**Keywords:** mathematics education; leadership; department; coordination; higher education; high school

---

## 1. Introduction

Leadership is one of the most sought after and reiterated improvement factors in the educational agendas of most countries [1]. A considerable number of research investigations has been carried out in compliance with the following guidelines of research promoted by the ISSPP (International Successful School Principalship Project) [2,3]; this research is located in the third stand, attending to the identification of personal qualities and generic professional competencies for effective school leaders.

In order to achieve the capacity to improve schools, with the support of leadership, the researchers return to the words of [4], who define it as "the conditions of the school that support teaching and learning, allow learning teachers and provide a means for the implementation of strategic actions to address the continuous improvement of the school" (p. 74).

Despite the great importance of this area of knowledge, relatively few studies on leadership are contextualized in mathematics. Most of them are linked to processes of instruction, improvement programs for student performance, or training programs dedicated to teachers of mathematics, to improve their professional performance through literacy processes or tools and methodological strategies, "alternatives". This research also arises from the demands imposed in this area of knowledge, when considered as instrumental, in international reports such as Programme for International Student Assessment (PISA), where the results are usually not particularly high in general. As a result, a more

professional role was provided for teachers, including the ability to lead [5]. In turn, this institution also highlights the difficulties mathematics teachers often have in teaching their subject due to their limited pedagogical training.

In an attempt to assess whether leadership skills equate to pedagogical limitations, they [4] conducted a study in 198 elementary schools in the Western United States. The purpose of their research was to corroborate the impact of leadership on student learning in the area of mathematics. Among the results, they confirmed that leadership had effects, albeit indirect, on student learning. They also determined "a perspective on school leadership and improvement as a process of mutual or reciprocal influence" (p. 83).

In short, it is a question of strengthening the quality of teaching in order to achieve the improvement of school learning. For this achievement, school leadership is one of the factors that most influences it, opening the way to favorable conditions where the professional learning of teachers takes place [6]. In fact, authors such as Miranda [7] identified leadership as the starter motor for lifelong learning, favoring the creation of new learning platforms, boosting the growth of professional learning communities whose aim is to improve professional teaching capital and give it the tools it needs to optimize its practice in the classroom.

## 2. Materials and Methods

### 2.1. Procedure

The review presented in this article is in line with the guidelines for carrying out a systematic review, included in the PRISMA (Preferred Reporting Items for Systematic reviews and Meta-Analyses) declaration of Liberati et al. [8]. Likewise, the standards defined by Fernández-Ríos and Buela-Casal [9] have been considered in the implementation process. The main objective of this work was to analyze all studies that address teacher leadership and coordination among mathematics teachers at different educational levels, with special emphasis on the impact on their professional development as teachers. For the search, the Web of Science database was used. This search was carried out during the months of September and October 2017, including the following keywords: "mathematic education"; "leadership"; "department"; "coordination"; "higher education"; "high school". In order to optimize the search, the Boolean operators "and" and "or" were used. The time range for the publication of these articles was then defined, from 2007–2017. In this time frame, 385 articles were found (Figure 1).

In this way, the population of the present analysis could be determined. Then, the search was refined considering only the articles included in the main collection of the Web of Science, within the research domain "social science" and the research area "education educational research". The sample of this study was the result of applying the following criteria of inclusion and exclusion: studies preferably empirical; the sample was constituted by teachers of mathematics; they considered the department of mathematics as a unit; that measures had been established to promote the professional development of teachers through leadership and/or other measures of internal change. Thus, works that were papers, doctoral theses, or articles that only had the abstract published were excluded.

The application of these inclusion criteria was carried out through a first reading of the title and summary of the study population; consecutively, a systematic reading of the full text of the articles was made. In this way, and with the application of conceptual, methodological, and statistical criteria, a total of 302 studies were eliminated.

With regard to data processing, a logical order comparison of the data was established, while all the information obtained was synthesized, resulting in a truthful and current study.

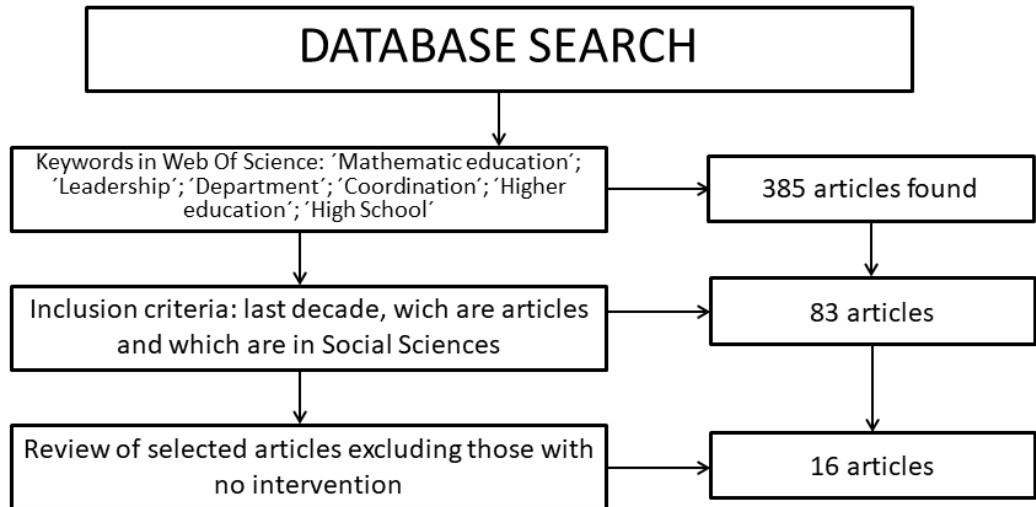

**Figure 1.** Flowchart of the article selection process.

### 2.2. Sample

According to the selection process previously described, the sample of this analysis was set at 385 articles, extracted from the Web of Science (WOS) (Figure 2). Once the inclusion criteria considered in the previous section had been applied, the sample resulting from this review was 16 scientific articles.

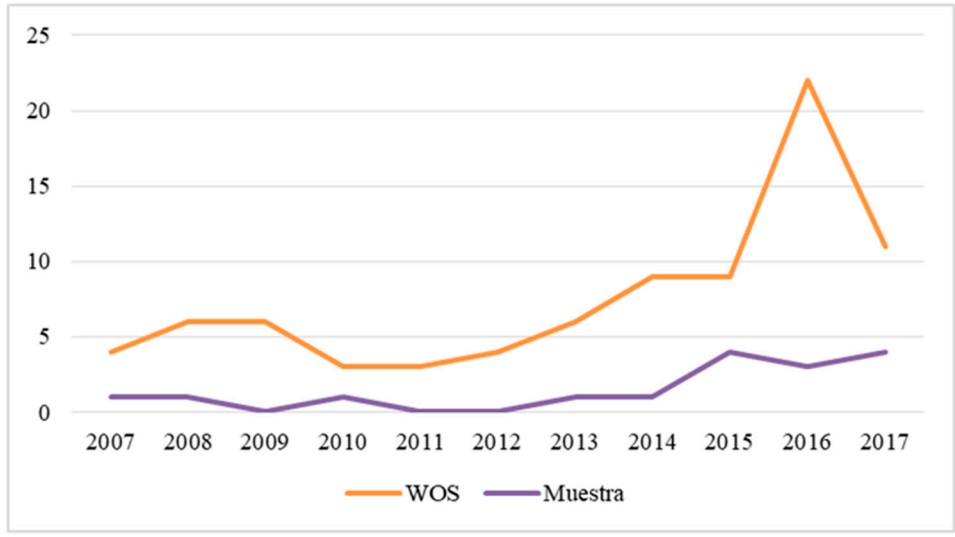

**Figure 2.** Distribution of publications from 2007–2017 in Web of Science.

## 3. Results

### Data from Studies Selected for Systematic Review

The 16 articles that made up the sample of this systematic review had a total sample participation of 6637, as shown in Table 1. In the extraction of the data, the following coding process was carried out: (1) author(s); (2) year of publication; (3) type of study; (4) population; (5) sample; and (6) the instrument used to examine the impact of leadership on the teaching practices of mathematics teachers.

**Table 1.** List of articles that made up the systematic review.

| Author | Year | Study Design * | Population * | Sample | Instrument * |
|---|---|---|---|---|---|
| Gibbins and Cob | 2017 | R | - | - | - |
| Kezar and Gehrke | 2017 | C | U | 4 | AD; I; O |
| Rigby et al. | 2017 | L | H | 4 | Instructional Quality Assessment and Instructional Bands (IQA); I |
| Sharma et al. | 2017 | L | U | 50 | ASERT Questionnaire; FG |
| Marco-Bujosa and Jurist Levi | 2016 | C | P | 5 | Comparative case study: I; O |
| Su and Bozeman | 2016 | C | U | 1832 | The 2010 Survey of Academic Chairs/Heads and a data-based assessment of research-doctorate programs in the United States |
| Yow and Lotter | 2016 | C | H | 16 | Q; Observation Protocol (RTOP) |
| Gómez et al. | 2015 | C | U | 3 | 12 Quantway® |
| Lew | 2015 | L | P, H | - | Pilot study MCPD |
| Maulana, Opdenakker and Den Brok | 2015 | L | H | 15 | Observation scheme LS |
| Summer and Sutherland | 2015 | R | K | - | - |
| Hopkins and Spillane | 2014 | C | P | 399 | School Staff Questionnaire; I |
| Blömeke and Klein | 2013 | C | P | 221 | TEDS-FU questionnaire |
| Heck and Moriyama | 2010 | L | P | 4000 | SEM multilevel |
| Sack | 2008 | L | H | 30 | O; D; RSP |
| Opdenakker and Van Damme | 2007 | C | H | 57 | Q; LISREL program |

\* R: Review; C: Cross-sectional; L: Longitudinal; U: University; H: High school; P: Primary education; K: Kindergarten; AD: Analysis of Documents; I: Interviews; O: Observations; Q: Questionnaires; FG: Focus Groups; D: Diary; RSP: Reflective Speech Participants; TEDS-FU: Teacher Education and Development Study: Follow Up; TDSM: Teacher Education and Development Study in Mathematics; SEM: multilevel structural equation model; LISREL: linear structural relationship model.

Regarding the educational level where the selected research was developed/implemented (see Table 2), it can be seen that the greatest number of research works were developed at the high school level (41.18%), followed by primary education (29.41%) and university (23.53%). Those carried out in kindergarten stand out for the small number (5.88%).

**Table 2.** Percentage and number of studies according to educational level.

| Educational Level * | Percentage | Study's Number |
|---|---|---|
| K | 5.88% | 1 |
| P | 29.41% | 5 |
| H | 41.18% | 7 |
| U | 23.53% | 4 |
| Total | 100% | 17 ** |

\* U: University; H: High school; P: Primary education; K: Kindergarten; ** One of the studies is contextualized in primary and secondary education.

Table 3 below shows the studies selected for this investigation, grouped according to the country in which they were carried out. It was found that most of the research on mathematics teachers and leadership was carried out in the USA. In the sample, we also found a review article that did not specify the demographic context.

**Table 3.** Percentage and number of studies according to country.

| Country | Percentage | Study Number |
|---------|-----------|--------------|
| USA | 62.5% | 10 |
| Belgium | 6.25% | 1 |
| Germany | 6.25% | 1 |
| South Korea | 6.25% | 1 |
| Australia | 6.25% | 1 |
| Indonesia | 6.25% | 1 |
| Unspecified | 6.25% | 1 |
| Total | 100% | 16 |

## 4. Discussion

The scientific articles analyzed in this work aimed to promote the professional development of teachers through internal measures. Some of these studies highlighted the implementation of leadership in schools, collaboration, or the creation of a climate of trust [10–17]. Other have carried out specific training programs to improve teaching [18–22]. Some have focused on the analysis of the figure of teachers based on educational policies and reforms promoted by the state [23,24]. Likewise, many of them have unified the science and mathematics faculty to analyze teacher leadership capacity and professional development [10,12,23–25]. However, a study was also found that had analyzed the instructional capacity and teaching practice of mathematics and English as a foreign language teacher together [21].

All of these investigations agreed to consider the promotion of the professional capacity of teachers as a priority aspect, bearing in mind that "high quality professional development should not stop with better teaching practice, but should strive to produce teacher leaders who share this growth with others as part of a systematic implementation of best practices" [12]. (p. 344). For this reason, many of the studies included in this analysis addressed communities of practice [23] and professional learning communities [12,13,16] as an important aspect for teachers to improve their professional performance. For example, we found the article by Kezar and Gehrke [23], who, from the literature found, determined a series of characteristics that undergraduate faculties should possess if they wanted to become authentic communities of practice. Among them, they stressed the importance of: (a) the implantation and development of leadership and its distribution; (b) a stable economic model that would allow its sustainability over time; (c) professional staff involved in the common project; (d) feedback processes and advisory mechanisms; (e) the need for research and evaluation; and (f) an articulated community strategy, where the requirements were discussed, as well as the implications for the future development of the community. Based on these considerations, it was possible to transform educational institutions into communities of practice from a non-formal perspective. An aspect that counteracts what happened is in the Yow and Lotter study [12], where spaces analogous to those of the professional learning communities were enabled for teachers to work together and learn from each other, through the exchange of practices. An important aspect to highlight of this study is that it was developed in high school. Its purpose was intended to empower mathematics and science teachers, opening the field to a more distributed leadership modality, through instructional trainers. Much of the success achieved in this study stemmed from the careful planning of the program. Once the bases were laid on what was to be done and the axes of intervention of the leadership instruction program were determined, the teachers were grouped so that they could plan, teach, and reflect together. This clustering allowed them to evolve their conception of teaching and learning processes, subordinated from a more traditional model towards one more in tune with current demands. In short, this study succeeded in establishing leadership among teachers thanks to the consideration that "just as effective professional development needs widespread support for teaching practice to change, teacher leadership must be supported and cultivated" (p. 343).

In parallel, several of the studies analyzed used the instrumental character of the area of mathematics. This allows designing instruction programs that would open bridges to the professionalization of the teachers of this subject, through collaboration with their peers or through external experts who would train them in instructional and pedagogical leadership. For instance, Rigby et al. [18] in their longitudinal study analyzed the expectations on the instruction programs of the participants, to determine to what extent the instruction received and the leadership had or did not impact their teaching practices, finding that they did not due to the lack of mathematical specificity in those programs. In order to counteract these results, the authors in the article demanded a greater endowment of significant resources to support management learning. Gómez et al. [19] opted for the use of innovative tools and methodologies as a way to grant greater professionalism to teachers. To this end, they subscribed to the professional learning model developed by [26], since in their opinion, it provided them with a reliable environment in which to analyze the experiences and narratives elaborated by the participants. Among the most significant findings of his study, it was found that a favorable position of the educational institution, in this case the faculty, towards innovation and the learning of teachers was determinant for the creation of knowledge structures, literacy of mathematical language, and improvement of teaching and learning processes; or the study carried out by [15], who advocated the importance of support for the teaching of mathematics and the creation of a work climate of trust on the part of management as a school leader, cataloguing them as a means to improve their professionalism. In turn, it influences the creation of knowledge structures, literacy of mathematical language, and improvement of teaching and learning processes. A further step was taken by [20], who linked the initial training of mathematics teachers with their subsequent professional development as teachers, in a study carried out in Korea. Among his evidence, this author found that Korean mathematics teachers held a favorable view of training programs, identifying them as a way to improve their professionalization as teachers, beyond the issuance of corresponding certificates. However, it is also contemplated that "several incentives given to teachers are a stimulus for the effective implementation of professional development programs in mathematics and contribute to the good performance of Korean students in mathematics achievement tests in PISA and TIMSS (Trends in International Mathematics and Science Study)" (p. 177). Therefore, it seems that professionalization also has an extrinsic motivational component. The work undertaken by [14] is also along these lines. These authors developed their investigation under the research of the educational school as a place of teacher training. To this end, a process was designed to support the learning of beginning teachers of mathematics and literacy in 24 primary schools in two school districts in the Midwest (USA). The findings highlighted the central role of leadership in creating professional development opportunities for teachers, especially beginners. In fact, it was also realized that "formal organizational structures within schools, particularly departments, as well as formal leadership positions, were important in advising beginning teachers seeking advice and information related to mathematics and literacy" (p. 331).

Other studies opted for awareness and knowledge of the educational policies [24], promoted by the government as a preliminary step to the implementation of leadership. The idea was for management and faculty to be aware of the unmet needs they had in the context in which they worked (and with families). As a consequence, they realized how leadership met those needs, based on professional development and collaboration. Similar, but considering the context and the profile of the students, was the research promoted by [16]. It presented the narrative of the professional experiences of a professional developer and thirty leading institute teachers within a training program. In order to resolve the conflicts inherent in mathematics instruction and leadership, collaborative work teams were proposed, culminating in professional support networks, obtaining greater success than expected. The research proposed by [17] could also be framed in this case study. In their study, the authors considered the impact that the school context has on students' academic results, relating it to school leadership in secondary education. As a result, it was determined that "the size of the school positively

affects school results and that its effect is mediated by the characteristics of school practice, such as the amount of cooperation among teachers, which affects the climate and school results" (p. 179).

Likewise, some studies have also been identified that have specifically studied the distributed leadership modality as a way of empowering teachers, obtaining an improvement in their professional capital. Among them, the one carried out by Sharma et al. [10] stands out, who put the focus on higher education and more specifically on the Science and Mathematics Network of Educators of the University of Australia (SaMnet). In this context, the authors focused on the innovative experiences of the participants. A strengthening of teacher relations and the promotion of a culture of collaboration as a consequence of distributed leadership were the most notable results. Indeed, there was little influence of the educational institution in the construction of initiatives for change and on the professionalism of teachers. Increasing concern about the need to become more involved staff were observed to achieve initiatives for internal change and strengthen distributed and pedagogical leadership in educational institutions. Related to that, Heck and Moriyama [22], based on a structural and multilevel model in primary schools, examined the impact of leadership on teaching practices and, therefore, on students' school results. Similar to Opdenakker and Van Damme [17], they found that contextual characteristics influence school outcomes, especially when there is room for a collaborative working context. However, they evidenced the impact of leadership on teaching practices and academic outcomes.

The study carried out by Marco-Bujosa and Levi [11] was based on the analysis of the traditional figure of the science teacher. It also presented a specialized model that gave it a more professional character in educational processes. The authors related leadership to a solid pedagogical program, through the distribution of leadership in the school.

Finally, most of the selected papers were empirical, except for the selected review articles [13,25]. The first responded to a review whose focus revolved around the challenges posed by the incorporation of leadership in the instructional processes of children's programs. Other studies collected information about problems at the educational level with the intention of responding to them. The emergence of curricular standards promotes good practices in the acquisition of mathematical pedagogical knowledge. Next, attention was given to the role of instructional leadership and its feasibility for inclusion in district program structures for transformation or consolidation into a professional community. Meanwhile, the second involved a systematic analysis that dealt with the measures to be implemented, as well as the fields of research for the improvement of the teaching practices of mathematics and science teachers. From the point of view of coaching, this article dealt with the professional learning to be carried out by the teacher's instructional trainer (and how it should guide the teacher), advocating that "instructional training is an important component of the professional development of teachers (...) it involves teachers who work with a more successful colleague, who exercises his main support in the work to improve teaching practices" (p. 411). Considering that it is necessary to promote the professional development of teachers in practice in order to improve their instructional work, a series of methods was presented to identify possible productive coaching activities, capable of fulfilling the "high expectations" that this area of knowledge possesses. After an analysis, the authors selected four types of activities as potentially productive in improving the professional development of teachers: (a) participate in the discipline, (b) examine the student's work, (c) analyze the classroom video, and (d) participate in the study of the lesson (p. 421).

## 5. Conclusions

Throughout the course of this analysis, there has been a profound change in the conception of teachers in general and of mathematics in particular, in terms of their professionalism and ways of managing teaching in the classroom. It is possible that one of the main reasons for this change has been the paradigm shift in teaching and learning processes in recent years, influenced, among others, by the incorporation of competencies into curricula. As a consequence of this, teachers are now required to take on the role of a leader, with the capacity to mobilize students in learning processes, while at the

same time building a climate of collaboration in the classroom, analogous to a learning space, as is the case in learning communities.

Another noteworthy aspect is that the leadership modality that tends to be attributed to the leading teacher is transformational or instructional due to the strong revaluation that accountability has acquired and the improvement at the level of quantifiable academic results.

In synthesis, the results found in this research are encouraging, since they indicate that leadership has positive effects on the professional development of teachers, as authors such as Darling-Hammond [6] have indicated. In this sense, it seems that the path to follow should focus on the line of empowering teachers, promoting their professional learning, through the construction of a work environment that motivates them to achieve this improvement.

At the same time, the need for more research of this nature also emerges if we want to achieve a real improvement in the processes of teaching and learning in the classroom.

Definitely, the formation of school principals as school leaders is an issue that should be further encouraged. Only through conviction of the effectiveness of distributed leadership in schools will teachers be able to adopt greater professionalism in their role. A good way to ensure the consolidation of the profile of leaders of teachers would be to design specific training programs. It should combine their role as teachers with their role as facilitators of processes. Consequently, it seems that these changes require more decentralized organizational structures in the school. This review article has set out the state of the issue in a general way in various contexts. In order to analyze it in greater depth, it would be desirable to conduct contextualized research in various countries. The first step would be to design a questionnaire that would provide a general overview of the state of distributed leadership in schools. We are currently along these lines in our research lab.

**Author Contributions:** I.G.-M., J.L.U.-J. and J.L.-B. conceived the hypothesis of this study. All authors contributed to review and lecture of the articles found. J.L.U.-J. and P.T. wrote the paper with significant input from I.G.-M. All authors read and approved the final manuscript.

**Funding:** This research received no external funding.

**Conflicts of Interest:** The authors declare no conflict of interest.

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
