# Peer review of "The Pedagogical Leadership of the Mathematics Faculty: A Systematic Review"

_education, doi:10.3390/educsci8040217_

Round 1
Reviewer 1 Report
This study investigates questions that have wide appeal to the field. However, there are some significant challenges within the paper. Much of what is currently included in the discussion sections, in reality, summaries of the articles included in the study. While the author synthesizes some of the arguments in the early part of the discussion, later parts simply move from one study to the next without connecting the studies in significant ways (see lines 202-252). Without the synthesis, the reader is left to wonder if the 16 articles included in the meta-study properly support the claims made in the conclusion. Finally, there are significant challenges in the English language constructions throughout.
Author Response
Firstly we would like to thank you for your time in reviewing our manuscript. We also thank you for your contributions to the improvement of the manuscript.
Changes have been made to the wording to facilitate the relationship between the different articles.
In the results section, the different articles have been grouped according to their most significant conclusions. See line 123-127; 137; 201-215; 216-232; 236.
Reviewer 2 Report
This paper requires review for proper tense, APA formatting, as well as clear determination in terms of being a review or actual study. It appears to be a meta analysis; however, presents as a review in some sections.Many of the sections that have been reported on are relevant to studies, as opposed to a review. I have made notes to further guide you, within the attached draft.

Author Response
Firstly we would like to thank you for your time in reviewing our manuscript. We also thank you for your contributions to the improvement of the manuscript.
All references have been reviewed and adapted to the Vancouver format.
The manuscript is a systematic review according to PRISMA recommendations. For this reason, the sections appearing in the manuscript have been included. In addition, the authors have carried out all the contributions they have made to us in their review.
Round 2
Reviewer 1 Report
The additions to the text has made the synthesis conducted more transparent and clear. There are still major issues in language and style; the manuscript will need detailed copy editing. In addition, I think the conclusion needs to be extended; given the results of this study, what are some next steps? What additional studies should be conducted or what policy recommendations might the authors suggest?
Author Response
Thank you very much for your time and contributions to improve the manuscript.
First of all, important editorial changes have been made. In addition, the conclusions section has been extended according to your suggestions.
Limitations of the study and possible future studies have also been added.
Once again, thank you very much for your contributions.